# Cutaneous Wound Healing and the Effects of Cannabidiol

**DOI:** 10.3390/ijms25137137

**Published:** 2024-06-28

**Authors:** Pearl Shah, Kathryne Holmes, Fairouz Chibane, Phillip Wang, Pablo Chagas, Evila Salles, Melanie Jones, Patrick Palines, Mohamad Masoumy, Babak Baban, Jack Yu

**Affiliations:** 1Department of Surgery, Medical College of Georgia, Augusta University, Augusta, GA 30912, USA; peshah@augusta.edu (P.S.); katholmes@augusta.edu (K.H.); fchibane@augusta.edu (F.C.); mjones27@augusta.edu (M.J.); 2Department of Oral Biology and Diagnostic Sciences, Dental College of Georgia, Augusta University, Augusta, GA 30912, USA; lewang@augusta.edu (P.W.); pchagas@augusta.edu (P.C.); esalles@augusta.edu (E.S.); 3School of Medicine, Louisiana State University Health Sciences, New Orleans, LA 70112, USA; ppalin@lsuhsc.edu (P.P.); mmasou@lsuhsc.edu (M.M.)

**Keywords:** wounds, wounding, wound care, cannabidiol, endocannabinoids, IL-33, alarmin

## Abstract

Cutaneous wounds, both acute and chronic, begin with loss of the integrity, and thus barrier function, of the skin. Surgery and trauma produce acute wounds. There are 22 million surgical procedures per year in the United States alone, based on data from the American College of Surgeons, resulting in a prevalence of 6.67%. Acute traumatic wounds requiring repair total 8 million per year, 2.42% or 24.2 per 1000. The cost of wound care is increasing; it approached USD 100 billion for just Medicare in 2018. This burden for wound care will continue to rise with population aging, the increase in metabolic syndrome, and more elective surgeries. To heal a wound, an orchestrated, evolutionarily conserved, and complex series of events involving cellular and molecular agents at the local and systemic levels are necessary. The principal factors of this important function include elements from the neurological, cardiovascular, immune, nutritional, and endocrine systems. The objectives of this review are to provide clinicians engaged in wound care and basic science researchers interested in wound healing with an updated synopsis from recent publications. We also present data from our primary investigations, testing the hypothesis that cannabidiol can alter cutaneous wound healing and documenting their effects in wild type (C57/BL6) and db/db mice (Type 2 Diabetes Mellitus, T2DM). The focus is on the potential roles of the endocannabinoid system, cannabidiol, and the important immune-regulatory wound cytokine IL-33, a member of the IL-1 family, and connective tissue growth factor, CTGF, due to their roles in both normal and abnormal wound healing. We found an initial delay in the rate of wound closure in B6 mice with CBD, but this difference disappeared with time. CBD decreased IL-33 + cells in B6 by 70% while nearly increasing CTGF + cells in db/db mice by two folds from 18.6% to 38.8% (*p* < 0.05) using a dorsal wound model. We review the current literature on normal and abnormal wound healing, and document effects of CBD in B6 and db/db dorsal cutaneous wounds. CBD may have some beneficial effects in diabetic wounds. We applied 6–mm circular punch to create standard size full-thickness dorsal wounds in B6 and db/db mice. The experimental group received CBD while the control group got only vehicle. The outcome measures were rate of wound closure, wound cells expressing IL-33 and CTGF, and ILC profiles. In B6, the initial rate of wound closure was slower but there was no delay in the time to final closure, and cells expressing IL-33 was significantly reduced. CTGF + cells were higher in db/bd wounds treated with CBD. These data support the potential use of CBD to improve diabetic cutaneous wound healing.

## 1. Introduction

Wounds and wounding are part of life, and treating wounds is in every physician’s job description. Some wounds are acute, others chronic. Some wounds are clean and result from surgeons’ incisions, the Class I or clean wounds according to the American College of Surgeons. Class I wounds heal better than dirty wounds [1]. However, even with pre-operative preparation using accepted disinfectants such as 70% ethanol, 4% chlorhexidine, or 10% povidone-iodine, clean wounds have an infection rate of around 1.5% [2]. Most traumatic wounds are not clean, contaminated by microbial agents and other materials present during injury. The total number of surgeries in the U S each year is about 64 million [3]. There are 8 million chronic wounds, defined as failure to reduce surface area by 50% per month, or with a duration greater than 3 months. The global prevalence of chronic wounds is 1.67% [4]. Breach of skin integrity underpins all cutaneous wounds, with mechanical disruption being the most common. When the stress pulling apart the skin exceeds the ultimate tensile strength (27.2 ± 9.3 MPa) of the dermis, the skin will fall apart, resulting in a cutaneous wound [5]. Tearing of the blood vessels in the papillary and reticular dermis and subcutaneous tissue causes bleeding necessitating hemostasis and initiates inflammation. Interference of blood flow causes regional ischemia, reducing blood supply to below the mean of 7 mL/100 g tissue/min. Skin, as an organ, receives 250 mL/min (5% of the total cardiac output) [6]. Other common, non-hemorrhagic, causes of local ischemia are external pressure (compressive stress) and thermal injury. When the external pressure exceeds the perfusion pressure, 1.33 kPa at the post-capillary venule level, local blood flow stops, with marked tissue damage at 32 kPa for as little as 15 min. This is particularly problematic over bony prominences. Fortunately, some capillaries remain at least partially open for some time, increasing the ischemia tolerance [7]. Prolonged and/or high pressure inevitably produces decubitus ulcers, especially when coupled with shear stress and excessive moisture. There are limits to how much and how many deviations from the nominal conditions (pressure, cold, or heat) living tissue can tolerate. The higher the magnitude of insult (I), the shorter the exposure time (t) required to cause cell death and tissue breakdown. Expressed mathematically, I × t = constant, or log I + log t = constant, similar to the production possibility curve (PPC). These constants exist because of underlying limits, in resources such as space, capital, labor, and time for PPC, and tolerable perturbation for cell viability [8].

Most surgically created wounds go on to heal, some develop surgical site infection and/or wound dehiscence and become chronic wounds. Many chronic wounds are not the result of surgical interventions. Pressure ulcers alone, for example, account for 2 million hospital admissions and 60,000 deaths per year [9]. These pressure-induced wounds are particularly prone in areas of bony prominence, such as the sacrococcygeal, ischial, and greater trochanteric areas, and increase the cost of care by USD 40,000 per hospital admission [10]. The total cost for the care of these pressure wounds was already USD 11 billion in 2011 [11]. There is an urgent need to understand and develop better treatments and prevention measures.

This brief review examines the cellular and molecular interplays of wounding and wound healing in normal and abnormal conditions with a special emphasis on the potential role of cannabidiol. The authors are a diverse group including surgeons, immunologists, cell biologists, geneticists, and biomedical entrepreneurs. We extracted from the vast peer-reviewed literature, supplemented with and supported by new data on cannabidiol, IL-33, and CTGF from our primary research. The intended target groups are clinical providers who care for patients with wounds and for researchers interested in the etiopathogenesis, normal and abnormal wound healing, prevention, and novel treatments of wounds. The introduction contains the four sections for updates on cellular and molecular events of normal wound healing, from inflammation, and coagulation, through proliferation to remodeling. It is followed by Results, Discussion, Materials and Methods, and Conclusion.
1.1Normal cutaneous wound healing
Early events: Coagulation and Innate Immune ResponsesNeuro-Endocrine Responses to InjuryProvisional Matrix and Transition to Collagen SynthesisDe-escalation, Remodeling, and Return to Homeostasis
1.2Conditions leading to and the mechanism of abnormal wound healing, both insufficient and excessive.
Chronic Wounds due to Vascular InsufficiencyGenetic Causes of Delays in Wound HealingExcess Scars: Keloids, Fibrosis, and ContracturesDelayed Healing due to Diabetes, Aging, and Obesity
1.3The effects of cannabidiol on normal cutaneous wounds.
The Chemistry of CannabidiolThe Endocannabinoid SystemIL-33 and Its Role in Tissue HomeostasisThe Effects of CBD on IL-33
1.4The effects of cannabidiol on abnormal cutaneous wounds.
CBD and Diabetic WoundsWound Healing in CB2-Knockout MiceThe Role of CTGF



**1.1** 
**Normal cutaneous wound healing**


***1.1A.*** 
**
*Early Events: Coagulation and Innate Immune Responses*
**


Injuries are unavoidable. Because of that, wound healing is essential for vertebrate survival and the wound healing process, highly complex and coordinated, is conserved during chordate evolution [12,13].

Some wounds are minor and may heal without medical or surgical care, while others may be life- or limb-threatening and require urgent surgical intervention. Immediately following an injury that causes the disruption of skin integrity, there is a loss of barrier function with a breach of vascular integrity (Figure 1). Within the epidermis, the basal cells lose contact with one another and the basement membrane. There is also disruption of the basement membrane. Yolk sac-derived, tissue-resident sentinel cells such as Langerhans cells within the stratum spinosum process microbial antigens and migrate to draining lymph nodes to present foreign antigens in an MHC II-restricted manner to naïve CD4+ T cells, initiating adaptive immunity [14]. Keratinocytes express toll-like receptors, (TLRs), and function as innate immune cells in addition to their barrier roles. Once their TLRs engage cognate ligands, keratinocytes produce many chemokines (CCL20, CCL27) and lymphokines (IL-8, IL-18) as well as TNF-α and interferon β [15]. Below the epidermis are the dermis (papillary and reticular layers), which imparts the tensile strength of the skin, and subcutaneous fat, which provides thermal insulation and energy storage. While the epidermis is avascular, the dermis contains capillaries, nerve endings, hair follicles with associated sweat and sebaceous glands, and pilo-erector muscles. There are also many other types of cells including fibroblasts, mast cells, macrophages, dendritic cells, and stem cells. Some epithelial cells and dermal fibroblasts disintegrate and die during the traumatic event, releasing their cytosolic and nuclear contents, such as Ca^2+^, K^+^, mitochondria, nucleic acids, histones, ATP, and cytokines, and growth factors without secretory signal sequences such as IL-33 and FGF-2. These normally sequestered intracellular elements are potent alarmins and bind to receptors (ST2, TLRs, for example) of the innate immune system to signal damage and danger [16]. This is a DAMP (Damage-Associated Molecular Pattern). A DAMP coupled with the detection of ligands of microbial origin such as mannose in close clusters, lipopolysaccharides (LPSs), and peptidoglycans elicits the most robust, rapid, and exponentially increasing inflammations. Mannose, through binding to a specific lectin (sugar-binding protein), the mannose-binding lectin, or MBL, activates the complement system by forming C3b via C3 convertase (C2b-C4B complex) [17,18]. Beginning with nitric oxide (NO) release, vessels dilate; neutrophils marginate and, with an increase in vascular permeability, extravasate by diapedesis. Pre-made vasoactive low molecular weight amines such as histamine and serotonin stored in granulocytes, especially mast cells, augment this. On the other hand, a fatty acid derivative released by platelets, thromboxane A2, synthesized from arachidonic acid, is an extremely potent vasoconstrictor that causes intense smooth muscle contraction [19]. Bleeding due to the disruption of blood vessels initiates multiple cascades of events. Circulating platelets adhere to injured endothelium and sub-endothelial connective tissue components such as type I collagen, and undergo rapid activation and aggregation, releasing their granular contents via exocytosis mediated by SNARE (soluble *N*-ethylmaleimide-sensitive factor attachment protein receptor), and forming primary hemostatic platelet plug. Von Willibrand factors (VWFs) released from the Weibel–Palade bodies within the endothelium help with platelet trapping. Platelets have 50 to 80 α-granules and release 300 soluble proteins, which are important mediators of coagulation (Factor V and VWF), inflammation (IL-8, CXCL1), and proliferation (FGF, VEGF, PDGF). In addition to this, they also release angiopoietin-2 which binds tightly to the A1 domain of VWF, promoting angiogenesis [20]. Dense granules, 150 nm in size, are smaller and fewer than α-granules, containing non-protein small bioactive amines (histamine, serotonin) and cations (mostly Ca^++^). Their release initiates inflammation and facilitates coagulation [21].

There are thirteen clotting factors, all but XIII are proteases and once activated by their own cleavage at the serine sites, catalyze the proteolysis of the next specific target factor. The initiation of the coagulation cascades is by tissue factor, Factor III (extrinsic pathway), and surface contact activation of Factor XII (intrinsic pathway). The liver produces four of these coagulation factors, II, VII, XI, and X, in the presence of Vitamin K to achieve gamma-carboxylation of the glutamate residues, allowing them to chelate Ca^++^. Both pathways converge to the final common pathway with the goal of converting soluble fibrinogen (Factor I) to insoluble fibrillary fibrin by the action of thrombin (Factor IIa), a serine protease, trapping red blood cells, platelets, and other circulating elements. This is the primary hemostatic plug. A transglutaminase rather than a serine protease, Factor XIII catalyzes the formation of the gamma-glutamyl-lysyl amide link to strengthen the clot against shear deformation [22].

Strategically located at two critical interphases, in the superficial dermis just deep to the epidermal basement membrane and just below the dermal–adipose junction, mast cells (MCs) participate in all three phases of wound healing. These marrow-derived special granulocytes make up 8% of all dermal cells and concentrate in the peri-vascular and peri-neural regions for expedited access to both. Immediately after injury, MCs undergo accelerated recruitment and activation through an autocatalytic loop formed by β-tryptase from MCs, which produces C3a-C5a, which recruit and activate MCs, releasing more β-tryptase. Overlapping with platelets, MCs degranulate and release vasoactive amines such as histamine and serotonin, and inflammatory mediators such as TNF-α. Since keratinocyte proliferation is premature and undesirable at this early time point after injury, histamine and heparin from MCs inhibit keratinocyte proliferation. By releasing MCP-1 (monocyte chemoattractant protein-1), MCs also recruit monocytes to the wound, polarizing them to M1. Importantly, MCs contribute to the antagonistic balance after the onset of thrombosis, coagulation, and inflammation by inhibiting the thrombin-induced conversion of fibrinogen to fibrin and releasing tissue-type plasminogen activator (t-PA). MCs also release IL-4, which is an anti-inflammatory lymphokine, and together with TGF-β can cause M1 conversion to M2. The close proximity of MC to fibroblasts also allows IL-4 to induce fibroblast proliferation. As wound healing progresses, keratinocytes proliferate and migrate along the denuded dermis, guided by integrin, fibronectin, and type V collagen. They come in close contact with MCs, setting up reciprocal stimulation: β-tryptase from MCs activates the PAR-2 receptor (protease-activated receptor, also known as GPR 11) on keratinocytes, which in turn releases stem cell factor, SCF, attracting more MCs. Three leukotrienes produced by MCs via arachidonic metabolism, LTB4, LTC4, and LTD4, cause keratinocytes to proliferate [23].

***1.1B.*** 
**
*Neuro-Endocrine Responses to Injury*
**


As the largest defensive barrier against the external environment, skin receives both afferent and efferent, as well as general and autonomic, innervations. To detect temperature changes (thermoreception), touch (mechanoception), and pain (nociception), the skin has three sets of general somatic afferent receptors. These primary sensory neurons synapse with secondary neurons in the dorsal root ganglia and cross over to the contralateral side before ascending to the thalamus. The internal capsule connects the thalamus to the somatosensory cortex. In addition to general somatic afferent nerve fibers, the skin has a rich network of autonomic innervations as well, providing secretomotor stimulations to the sweat glands and innervation to the pilo-erector (by cholinergic postganglionic sympathetic efferent neurons) [24]. Cutaneous injuries cause stress responses involving the hypothalamus–pituitary–adrenal (HPA) axis, releasing corticotrophin-releasing hormone (CRH). In addition to its central effects, CRH also binds to the peripheral cognate receptor, CRH-R1, on keratinocytes and fibroblasts in the skin, leading them to secrete adrenocorticotropic hormone, ACTH. In a paracrine manner, ACTH in turn, through melanocortin receptors, causes keratinocytes and fibroblasts to release IL-1, cortisol, and corticosterone. Keratinocytes, but not fibroblasts, also release three additional interleukins: IL-10, IL-12, and IL-18. IL-12 induces pro-inflammatory Th1 while IL-10 indices anti-inflammatory Th2. This is another example of the delicate antagonistic balance [25].

For many decades, physicians have observed massive elevations in serum norepinephrine levels following severe injuries. Recent data indicate such catechol rise interferes with immune cell trafficking and decreases lymphatic efflux. Alterations in aquaporins through the action of norepinephrine are a possible mechanism [26]. This reduction in lymphatic outflow contributes to higher neutrophil and macrophage counts in cutaneous wounds. During the proliferative phase, norepinephrine reduces angiogenesis. The total time to re-epithelization stays the same but with a two-fold initial delay [27,28].

Injuries are painful due to nociception and the release of biochemicals such as substance P and bradykinin. Normal afferent inputs, such as touch, warmth, and cold, become pain through activation of a special kind of Ca^2+^ channel, transient receptor potential vanilloid 1 (TRPV-1), when the magnitude of stimulations exceeds thresholds that may cause cell death and tissue damage. The nociceptive system thus functions as an alarm mechanism to quickly detect potentially damaging conditions. Its actions rely on nociceptors, which are specialized receptors that preferentially detect noxious stimuli and are activated by touch, warmth, cold, and/or chemicals. Nociceptors are evolutionarily conserved and found in all animal species, with evidence that suggests consistent signaling mechanisms across species [29]. Nociceptors, transmitting via small, myelinated A-delta fibers and small unmyelinated C fibers, process the modality, intensity, and duration of the stimulus. The signals are carried from the periphery to the brain via dorsal root ganglia and the spinothalamic tracts or the trigeminal nerve to the Gasserian ganglion. Pain perception then occurs in the somatosensory cortex with the thalamus serving as the key relay. The brain also exerts descending control over pain, reducing nociception when appropriate. Dysregulation of this process is implicated in the pathology of chronic pain [30].

Nociceptors also have a role in the local transmission of information in the periphery. The skin’s C-fiber nociceptive terminals contain vesicles replete with neuropeptides, notably substance P and calcitonin gene-related peptide (CGRP). Substance P is secreted by neurons and binds to the neurokinin 1 receptor to mediate its effects in the nociceptive pathway [31]. The release of substance P increases sensitivity to pain and pro-inflammatory cytokine release [32]. CGRP and its receptors are expressed in nociceptive pathways integral to the pathogenesis of migraine headaches [33]. Upon nociceptor stimulation, the release of these neuropeptides causes inflammatory effects in the skin. Redness and swelling are due to local vasodilation and plasma extravasation, respectively [29]. Damage to nociceptors after injury can cause a decrease in stimulus threshold, thus causing an increased pain sensation in a wounded area and subsequent protection of the area, promoting healing [34]. This phenomenon is called sensitization, and it explains the mechanisms behind both allodynia, where nociceptors are activated by a normally inoffensive stimulus, and hyperalgesia, where nociceptors are activated more intensely by painful stimuli. Both play important roles in complex regional pain syndromes.

***1.1C.*** 
**
*Provisional Matrix and Transition to Collagen Synthesis*
**


After the inflammatory phase, with overlaps, the wound enters the proliferative phase, beginning on day 3 following injury, and starts to produce provisional matrices made up of fibronectin, glycosaminoglycans, and mucopolysaccharides. Their functional roles are to quickly occupy the wound, replace the fibrin clot, and prevent continued exposure to the external environment. They also guide the laying down of their replacements—the collagenous matrix. There are at least seven potential biopolymers that make up the provisional matrix, including hyaluronic acid, chondroitin, chondroitin sulfate, dermatan, dermatan sulfate, heparan, and heparan sulfate. The sulfated proteoglycans are more important in converting isotropic, non-fibrillar ground substances to a more fibrillar architecture. The provisional matrix has little compressive or tensile strength. For that, the wound needs collagen, especially type I, synthesized by fibroblasts. Collagen makes up ⅓ of the total protein in our body and is very stiff. The Young’s modulus of type I collagen is at least 100 MPa [35].

Fibroblasts, differentiated from local stem cell pools such as small pericytes, appear in the wound by the second to fourth day, with detectable collagen synthesis by day 4 to day 6 (Figure 2). Histamine and tryptase from MCs stimulate fibroblast synthesis and the release of SCFs and FGFs. SCFs recruit more MCs and induce MCP-1. MCP-1 increases transcription of *α*1(*I*) of collagen type I. By the end of week 1, the collagen production rapidly increases, until the middle part of week 3 (day 17), with minimum increase after that, returning to nominal levels by the end of week 6 in the normal healing process. Collagen undergoes extensive post-translational modifications, involving hydroxylation of proline and lysine residues and the formation of covalent cross-links. Proline residue, or prolyl, hydroxylation requires proline-4-hydroxylase, a member of the alpha-ketoglutarate-dependent hydroxylases located in the endoplasmic reticulum, with ferrous ion, Fe^2+^, as a cofactor and vitamin C as an electron donor to prevent the oxidation of iron from the ferrous to ferric state [36]. Molecular oxygen is the source of oxygen in proline hydroxylation. From these two crucial but opposing roles, the importance of tightly regulated oxygen levels is clear: normal healing cannot occur with too much or too little oxygen. The oxidative deamination of the epsilon amino group of the lysine residues forms aldehydes, which is another important site for covalent cross-links while regenerating the aldehyde group. Lysyl amino oxidase, with copper as a cofactor, catalyzes this reaction. 4-hydroxyproline is exclusively present in collagen, making up 9% of the molecule with proline contributing 12% and glycine 35% [37].

In addition to angiopoietin 2 released by the activated platelets, MCs release angiopoietin 1 and 7 other pro-angiogenesis mediators: VEGF, TGF-b, FHF-2, NGF, IL-4, PDGF, and IL-8. The new angiogenesis assures that the healing wound has sufficient nutrients and oxygen.

***1.1D.*** 
**
*De-escalation, Remodeling, and Return to Homeostasis*
**


Antagonistic balance is the sine qua non of homeostasis. This holds true for all phases of wound healing including coagulation and inflammation. There are elaborate fibrinolytic (kallikrein, plasmin) and antithrombotic measures (Protein C and AT-III) to lyse the clot and prevent undesirable coagulation. There are also potent anti-inflammatory lymphokines (IL-4, IL-10) and growth factors (TGF-beat) to polarize the pro-inflammatory M1 to anti-inflammatory M2 [38]. During the proliferative stage, collagen deposition typically peaks by the third week post-injury, resulting in scar tissue that provides temporary tensile strength to the wound. The primary component of scar tissue is fibrillar type I collagen. The collagen fibers of scar tissue are characterized by disorganized fibers in contrast to the reticular pattern found in the native dermis. After sufficient scar or granulation tissue is deposited over the wound bed, wound edge epithelium must then expand and cover the denuded tissue to complete the healing process [39]. Scar formation is the ultimate outcome of wound repair in children and adults. Following revascularization and epithelialization, the wound may now be excluded from the external environment. However, wound healing is not yet complete. Scar tissue subsequently undergoes a process of remodeling that can recover up to 80% of baseline tensile strength. For this to occur, it is critical to de-escalate the heightened humoral, inflammatory, and adaptive immune responses to avoid the overactive formation of scars.

Scar remodeling occurs over the course of a year and includes the reorientation of collagen fibers to become progressively more anisotropic, aligning themselves along the direction of external forces to better resist tensile strain. Furthermore, other processes including vascular ingrowth, stimulation of myofibrils, and regeneration of nerves [40] act to restore homeostasis. This is accomplished by the actions of several key cell types, most notably type 2 macrophages (M2) and T-regulatory lymphocytes (Treg), which function to suppress many of the activated responses of resident cells and clear excess ECM [41,42]. During remodeling, macrophages transform from type 1 macrophage (M1) to the reparative M2 phenotype. M2 serves to physically clean the wound bed by phagocytosing neutrophils, bacteria, and cellular debris, while also chemically, via the synthesis of anti-inflammatory mediators, degradative enzymes such as matrix metalloproteinases (MMPs), initiating myofibroblast proliferation and stimulating angiogenesis [43,44].

During the remodeling process, wound contracture occurs through the action of myofibroblasts. Myofibroblasts exert their contractile forces by focal adhesion complexes that link the intracellular elements such as the cytoskeleton to the ECM. Early scars appear pinkish red due to their dense capillary networks; regression of these capillaries allows scars to lighten significantly over time. Ultimately, the formation of a mature scar marks the end of the remodeling phase (Figure 3). In addition to a collagen pattern distinctly different from native skin, epidermal appendages such as pilosebaceous complexes are lacking in mature scars [45].

**1.2** 
**Abnormal cutaneous wound healing**


***1.2A.*** 
**
*Chronic Wounds due to Vascular Insufficiency*
**


Most wounds heal uneventfully within 90 days of injury progressing through the three phases of wound healing. However, in 1–2% of wounds, the process lingers at the inflammatory phase and fails to move beyond that. The result is an open, chronic wound. There are many reasons for wound healing to go awry, often delayed as in chronic wounds but at times excessive, as in hypertrophic scars and keloids. For chronic wounds, it is important to examine the “whole” patient before examining the “hole” in the patient (Figure 4). Causal relationships in biology and medicine are often hard to establish definitively, due to their multifactorial and context-dependent nature. These are the defining characteristics of complex adaptive systems [46,47]. It remains always true that insufficient nutrient and oxygen delivery to the tissue will result in cell death and tissue breakdown, as life occurs far from thermodynamic equilibrium. Since the vascular system supplies all the required substrates (lower entropy) and removes all the wastes (higher entropy), the compromise of this mass transfer apparatus is always a part of the chronic wound. Such compromise may be at the macrovascular level or, as in the case of pressure ulcers, at the level of the capillaries.

Other contributing factors include the nature of the initial injury, such as the extent of tissue destruction, contamination, patients’ age, and general pre-injury medical conditions. Using the NCPIGGN mnemonic, healthcare providers in any setting can quickly, and completely, obtain an assessment of the overall health of the patient. NCPIGGN stands for neurologic, cardiac, pulmonary, infectious diseases, gastrointestinal, genitourinary, and nutrition. Cellular dysfunctions, qualitative, quantitative, or both, underpin most chronic wounds stuck in the inflammatory phase. There is persistence of M1 rather than the M2 phenotype; neutrophils remain in the wound; fibroblasts, though present, are senescent. Mast cells are too many and contain excess granules with pro-inflammatory mediators such as TNF-α. Microbial colonization and infection promote a state of high MMP levels, degrading the growth factors necessary for normal healing [48].

Chronic venous insufficiency is a cause of venous leg ulcers, the most common type of leg ulcers. These are frequently located in the gaiter region, an area between the midcalf and the ankle. Venous ulcers tend to recur and require frequent healthcare provider visits to dress the wound. This poses a significant cost burden to the patient while decreasing the quality of life [49]. Normally, retrograde venous blood flow is prevented by one-way valves that open toward the deep veins. In patients with venous insufficiency, this process is impaired, causing inappropriate transmission of blood from deep veins to superficial veins [50]. This results in varicose veins where the normal vein wall architecture is disrupted and “venous hypertension” which can lead to ulceration (Figure 5).

The pathophysiology of venous ulceration is complex but has become more fully understood over the last two decades. Current theories suggest that venous hypertension elicits a state of chronic inflammation in which the constant recruitment of white blood cells leads to tissue injury [51]. Increased pressures in the vessels lead to extravasation of inflammatory agents, such as fibrinogen and iron from red blood cell degradation [52]. The inflammatory signals result in the recruitment of macrophages and mast cells, which release transforming growth factor beta-1 (TGF-β1) and histamine, respectively. TGF-β1 inhibits fibroblast proliferation, impairing collagen production. Lastly, due to iron overload, macrophages fail to polarize to the M2 phenotype required for extracellular matrix deposition and therefore, proper wound healing [51]. At the molecular level, the regulation of tissue oxygen content is obviously critical. Hypoxia-inducible factors (HIFs) are integral to this important regulatory network, functioning as oxygen sensors. Prolyl hydroxylase, responsible for adding –OH to carbon 4 of the imino ring of proline residues in type I collagen using molecular oxygen, is a member of the 2-oxoglutarate, also known as α-ketoglutarate, -dependent dioxygenases family. The prolyl hydroxylase domain (PHD)-containing proteins also hydroxylate proline residues in HIF-1α and HIF-2α in the presence of Fe^2+^. Chronic open wounds with exposure to atmospheric O_2_ cause excessive Fe^3+^ production and interfere with prolyl hydroxylase and HIF function [53]. Importantly, iron through a Fenton reaction in an oxidative environment, generates an extremely reactive hydroxyl radical, OH., which can cause extensive oxidative damage to the organelles [54].

***1.2B.*** 
**
*Genetic Causes of Delays in Wound Healing*
**


Because of the complex nature of wound healing with the requirement of coordinated ECM productions and turnover, many mutations in the enzymes needed in such matrix production or the matrices themselves can, at least in theory, affect wound healing. These mutations often produce observable phenotypes related to connective tissue disorders such as dystrophic epidermolysis bullosa and Ehlers–Danlos syndrome. But they may have relatively mild effects under normal conditions because if the disruption of the ECM due to a certain mutation is severe enough to cause structural defects, it is likely to be embryonically lethal. As such, the mutation will not propagate in the population. Here we discuss mutations in two families of genes, one related to the provisional matrices and the second to the collagenous matrices, which overlap as the latter requires the former. Since the sequencing of the human genome and wider application of genetic tests such as the Genome-Wide Association Study (GWAS) and Whole-Exome Sequencing (WES), many variations from the reference sequence have been reported. Many of these variations have no documented significance and are classified as VUSs: variations of unknown significance. For those variations limited to one nucleotide that occurs in more than 1% of the population, their designation is SNP for single-nucleotide polymorphism. Mutations, on the other hand, can be deletions, duplications, and insertions, which cause frameshift, as well as single-base substitutions as in SNP, but with a frequency less than 1% in the population [55].

All seven provisional matrices require the transfer of monosaccharides to form sugar side chains which then attach to the protein core. Dermatan, chondroitin, and heparan undergo sulfation by post-translational modifications. Galactosyltransferase 1 first transfers galactose to xylose in the proteoglycans’ linkage region, a crucial step in the production of heparan sulfate. The arg270cys mutation in *B4GALT7*, a gene coding galactosyltransferase 1, has been documented in delayed wound healing in addition to accelerated aging and Ehlers–Danlos syndrome [56]. Other mutations, unrelated to galactosyltransferase can affect non-collagenous matrix production. A frameshift mutation in ATP6V0A2, a vacuolar H^+^ pump, for example, is associated with cutis laxa and delayed wound healing [57].

Seven mutations in *COL1A* have been identified, two in *COL1A1* and four in *COL1A2*. Four of the seven suffered delays in wound healing. In addition to healing issues, these individuals show hypermobility, easy bruising, translucent skin, and hypotonia. Both *COL1A1* mutations, gly188asp and gly203cys, are related to a disturbance in wound healing [58].

Despite the vast number of mutations and SNPs in ECM genes, delays in wound healing are rarely due to genetic causes alone. In the absence of other findings, it is nearly impossible that isolated wound healing failure is the only manifestation. Clinicians should therefore work up other, non-genetic causes before attributing the problem to genes.

***1.2C.*** 
**
*Excess Scars: Keloids, Fibrosis, and Contractures*
**


Even with clean surgical wounds, a hypertrophic scar, defined as a raised scar remaining within the original wound base, develops in 30% of the cases. When raised parts of the scar exceed the original base, it is a keloid. Keloids occur less frequently in the general population, about 10%, but are more difficult to treat due to a high recurrence rate (Figure 6). Both environmental and genetic factors influence the development of excess cicatrix. Increased tension, infections, darker skin, foreign bodies, and genetic predisposition are the common culprits. Almost all these excessive fibroproliferative processes eventually reach a steady state with collagen production balanced by collagen degradation. The problem is that scar mass accumulation has already occurred during the period when collagen synthesis far exceeded its degradation [59].

A recent study examining the serum and wound levels of endocannabinoids in 50 women undergoing body contouring aesthetic surgery found a better correlation between systemic and skin endocannabinoids (r = 0.38, *p* = 0.015) in those without hypertrophic scars. Importantly, the arachidonoylethanolamine level was significantly lower in the hypertrophic scar group (*n* = 10) when compared to the group with normal healing (*n* = 40), 0.77 ± 0.12 ng/g vs. 1.15 ± 0.15 ng/g, *p* < 0.001. The arachidonoylglycerol levels peaked by day 5 while returning to a steady baseline by 3 months [60].

The histology of keloids shows dense, disorganized bundles of collage with an increase in fibroblasts. Immunohistochemistry of keloids reveals the presence of CD34−/α-SMA+/p16+ cells, in contrast to CD34+/α-SMA-p16- cells in the dermis of normal adjacent skin. Due to myofibroblasts, wounds of critical areas, such as the first webspace of the hand, the submandibular area of the neck, elbow, knee, axilla, and hip, undergo contraction and result in contractures which impede normal function by restricting both active and passive range of motion. Fibrosis is a major problem preventing full recovery after musculoskeletal surgeries. In 4% of the patients receiving breast implants, this manifests as contracture surrounding the implant [61].

Intestinal adhesions between loops of bowels are the dreaded consequence of abdominal injuries from either trauma or surgery, occurring in 10% of the cases. This highly undesirable intra-abdominal scarring can cause infertility, small bowel obstruction, internal hernia, twisting, and strangulation of the intestines.

Even though this review focuses on normal and abnormal healing of the cutaneous wounds, there is a massive amount of data on disease burden due to excessive “healing” resulting in fibrosis. Liver cirrhosis, pulmonary fibrosis, glomerulosclerosis of the kidney, and post-infarction myocardial scarring, in addition to the above-mentioned intra-abdominal adhesions, are examples of such and all have one factor in common: IL-33.

***1.2D.*** 
**
*Delayed Healing Due to Diabetes, Aging, and Obesity*
**


The prevalence of obesity, defined as BMI > 30 kg/m^2^, continues to increase, reaching almost 42% in 2017 [61]. The same is true for diabetes, with 38 million people having the disease and 97 million people having pre-diabetes in the U.S. [62]. Although the cause of such a rise in the population level is complex and certainly multifactorial, the effects on individual health are unequivocal. Human genetics and physiology evolved and adapted millions of years ago, which are inappropriate for the current environment, where there are abundant high-caloric foods and little need for physical exertion for, and the avoidance of, predation. Simultaneously, there is aging of the U. S. population, with 16.5% at or older than 65 years in 2020 [63]. Though not independent variables, each of the above three groups poses special challenges in managing their wounds.

The cause of obesity is multifactorial. Obese individuals are more likely to have metabolic issues such as hyperglycemia, T2DM, and hyperlipidemia, among others. Isolated obesity without other compounding factors can delay wound healing simply by the difficulties inherent in the physical form such as the folds and intertrigo that result from them, especially when the BMI is above 40 kg/m^2^. Though not as active as skeletal muscles, adipose tissue has the same cytoplasm-to-capillary ratio. This puts extra demands on the cardiovascular system. However, as BMI worsens, the vascular density fails to keep up with fat accumulation resulting in relative regional ischemia, impairing healing. At the ECM level, there is a decrease in elastin and an increase in types V and VI collagen. At the cellular level, there is an M1 dominance with more iNKT cells (invariant natural killer T cells). There is an overall much more pro-inflammatory presence with, paradoxically, relative hypoxia and elevated oxidative stress [64]. Not only is the quantity of food intake important, but the quality of the diet is also a contributing variable. Fructose, a hexose with a ketone that is sweeter than glucose, is an aldose. Fructose creates more cravings. Industrial processes such as HFCS 55 split the sucrose in corn syrup, a disaccharide, into two monosaccharides, producing 55% fructose to make it much tastier to the consumers. The problem is ketose such as fructose suppresses glucose metabolism and, through TCA cycles, combines with acetoacetates, producing a large number of fatty acids by de novo lipogenesis and heightening inflammation [65,66].

For diabetes mellitus, type 2 (T2DM) accounts for most of the increase, with only 7% of diabetic adults having T1DM. Whether insulin insensitivity (T2DM) or insufficiency (T1DM), the sine qua non is hyperglycemia. With persistent high glucose levels exceeding the processing capacity of hexose kinase, aldose reductase converts the excess glucose to sorbitol using the polyol pathway. This is particularly problematic for the peripheral nerves, which contain aldose reductase. Sorbitol is a strong osmolar agent and causes swelling of the cells. Constrained by the peri- and epineurium, axonal transmission, especially the afferent fibers, becomes impaired. This numbness due to diabetic neuropathy increases the risk of injury [67].

Hyperglycemia causes non-enzymatic glycation of proteins and lipoproteins through the Amadori reaction followed by condensation with lysine and arginine residues, forming advanced glycation end products, or AGEs. As oxidants, these glycosylated adducts abnormally cross-link basement membrane proteins, including type IV collagen, and induce endothelial dysfunction, leading to atherogenesis and angiopathy at both micro- and macro-vascular levels. Such pathology interferes with the autoregulation of blood flow.

At the ultrastructural, genetic, and epigenetic levels, diabetic wounds show prolonged inflammatory response due to many reasons, including the persistence of the M1 phenotype. One mechanism is extended *NF-KB*-mediated transcription due to elevated levels of Jumonji domain-containing protein D3 (JMJD3), which opens the promoter site by demethylating H3K27Me3- lysine residue in position 27 of histone 3. Using single-cell RNA sequencing of human and transgenic *Jmjd3*-deficient murine cells from diabetic and non-diabetic wounds, CGAS-STING appears to be a downstream effector, implicating the activation of innate immunity by intracellular DNA binding in diabetic wounds [68].

Because of all the above, diabetic wound healing does not progress normally and tends to remain in a chronic inflammatory state. Combining the micro- and macro-level abnormalities, the lifetime risk of developing foot ulcers in diabetic patients is one in three, with many eventually requiring amputations [69].

**1.3** 
**Effects of cannabidiol on normal cutaneous wounds**


***1.3A.*** 
**
*The Chemistry of Cannabidiol*
**


Cannabidiol, or CBD, is one of at least 85 active cannabinoids produced by *Cannabis sativa*, a member of the Cannabaceae family. Similar to catechol, it is a modified benzene diol with a molecular weight of 314.5 Dalton. Two functional groups, a cyclohexene and a pentane, attach in the para- positions (carbons 2 and 5) of the benzene diol. The hydroxyl groups of the “diol” are themselves in the meta-, or 1, 3, positions. The functional group attached to carbon 2 is a modified cyclohexene with methyl and isopropene: 3-methyl-6-(1-methylethenyl)-2- cyclohexene. Attached to carbon 5 is an aliphatic tail of 5 carbons, similar to the pentyl tail of arachidonic acid, pentothal, many prostaglandins, and endocannabinoids (Figure 7). The putative receptors for CBD are CB1 and CB2, with the former expressed more in the CNS and the latter having a more general, peripheral distribution. Even though the CB1 level is high in the hippocampus and amygdala, it is the most widely distributed GPCR in the human body.

CBD is essentially not soluble in water, only 12.6 mg/L. Its solubility in a non-aqueous solvent is 5000 times higher, 60 mg/mL in DMSO.

***1.3B.*** 
**
*The Endocannabinoid System*
**


Both CB1 and CB2 are the G-protein coupled receptors (GPCRs) for endogenous cannabinoids or endocannabinoids (eCBs). Widely distributed in the CNS, especially in GABA-nergic interneurons, CB1 is also present in some peripheral tissues and organs such as fat, liver, and skin [70]. CB2 expression is less central and more in immune cells. Even though several bioactive lipids can bind with CB1 and CB2, there are two major eCBs: 2-arachidonoylglycerol (2-AG) and *N*-arachidonoylethanolamine (anandamide, or AEA); both are lipid ligands, modified from arachidonic acid. 2-AG and anandamide retain the key features of their parental precursor: arachidonic acid: icosa-cis 5,8,11,14-tetraenoic acid with a pentyl tail (Figure 8) [55]. The synthetic pathways are different for the two eCBs, with diacylglycerol lipase for 2-AG and *N*-arachidonoyl phosphatidylethanolamine phospholipase D for AEA. The degradation of 2-AG by monoacylglycerol (MAG) lipase returns it to glycerol and arachidonic acid, while the degradation of AEA by fatty acid amide hydrolase (FAAH) returns it to ethanolamine and arachidonic acid. Murine models with reduced FAAH function show high pain tolerance (hypoalgesia) and, interestingly, “accelerated wound healing”. A woman with a microdeletion of the *FAAH* gene complex, documented by WES, had markedly elevated levels of eCBs: twice the normal serum level of anandamide at 2 pmol/mL, oleoylethanolamine (17 pmol/mL, 4× normal), and palmitoylethanolamide (110 pmol/L, 3× normal). She demonstrated profound hypoalgesia, reporting 0/10 in pain, following hip replacement and wrist surgeries [71].

In addition to CB1 and CB2, 2-AG and anandamide can also bind to transient receptor potential vanilloid (TRPV) channels and peroxisome proliferator-activated receptors gamma (PPARγ). The signaling downstream of eCB-receptor binding includes inhibition of voltage-gated ion channels and adenylate cyclase activation by cAMP, among others. Due to the large size of GPCRs, allosteric modifications from ensemble effects are common. Further complicating the pleiotropic effects is their propensity for heterodimerization and multimerization [72]. Human keratinocytes express many eCB receptors including CB1 and CB2. Even at 1 μM concentration, AEA could inhibit keratinocyte differentiation in a CB1-dependent manner. At 30 μM concentration, AEA inhibited keratinocyte proliferation while also inducing its apoptosis [72].

Through the above mechanisms, endocannabinoids maintain epithelial tissue homeostasis. They also regulate inflammation, as evidenced by studies using CB1−/− and CB2−/− mice. CB2 deletion increased inflammation (higher IL-6 and TNF-α) but with no delays in wound healing. CB1 deletion, on the other hand, did prolong wound healing time with higher levels of monocyte chemoattractant [73].

Extensive research over the past several decades has investigated the effects of CBD on cutaneous wound healing, with some performed using human subjects [74]. Most found CB2 to be the key, reducing the pro-inflammatory M1 macrophages, acute phase reactants, and enhanced epithelization. Due to its high lipid solubility, CBD delivery is complicated, requiring emulsification into 150 nm to 300 nm microdroplets. A commonly used anesthetic agent, di-isopropyl phenol, under the generic name of propofol, has the same lipid solubility issues [75]. Because of this, topical applications have become more popular. Agonist ligand for CB2, but not CBD, applied to murine cutaneous wounds has shown efficacy in reducing inflammation and fibrosis [76].

Cultured keratinocytes and fibroblasts, stimulated by TNF-α, reduced the expression of pro-inflammatory genes when treated with CBD [77]. Other CB2 agonists such as JWH015 showed similar effects, stimulated not with TNF but with LPS [78]. An interesting study in 2011 using a 1.5 cm dorsal incision in mice showed that CB2 is normally expressed in the dermis. Following injury, CB2 levels increased until days 3–5 in M1 and myofibroblasts. Importantly, CB2 agonists decreased fibrosis as in the in vitro studies [79].

***1.3C.*** 
**
*IL-33 and Its Role in Tissue Homeostasis*
**


A receptor without known ligands is an orphan receptor. ST2, first discovered in 1989 was an orphan receptor, with both transmembrane and soluble forms, ST2L and sST2, respectively. Following the completion of the Human Genome Project on April 14, 2003, the sequence data allowed for a computerized search for potential ST2 ligands. Such efforts led to the discovery of a new member of the IL-1 family in 2005, receiving the name IL-33. Human IL-33 has 270 amino acid residues and stimulates type 2 immune responses through the polarization of T cells to T_2_H [80].

Mapped to chromosome 9p24.1, the human *IL-33* gene has 16 kb in 7 exons with a large 25.8 kb Intron 1 and 8.9 kb Intron 2. IL-33 is a member of the IL-1 superfamily of alarmins, or DAMP (Damage-Associated Molecular Pattern), and it sends danger signals to alert and activate the innate immune system to respond to tissue damages that have occurred. Stored in the nuclei, IL-33 is particularly relevant in regulating immune responses to injury. Injury causes cell death. With cell death by necrosis or necroptosis, nuclear IL-33 enters the extracellular space and engages its cognate receptor ST2 (Suppressor Tumorigenesis 2). The ensuing signal transduction, acting through MyD88 to IRAK1, a 4-TRAF6 complex, causes activation of MAPK and IkB degradation after phosphorylation, which activates AP-1 and NF-kb. The results of the activation of these transcription factors are highly pro-inflammatory. However, within the nucleus of intact cells, IL-33, with its HLH motif, can bind to DNA. The *IL-33 N*-terminus also codes for proteins with chromatin-binding motifs, strengthening its ability to regulate pro-inflammatory gene expression [81].

In addition to its role in inflammation, IL-33 binds to smooth muscle cells and elicits Ca^2+^ influx, resulting in contraction [82]. This is an important mechanism for expelling parasitic worms that have invaded the intestinal epithelia, which cause barrier damage and release of IL-33. Recent animal experiments confirm that IL-33 enhances wound healing and pro-fibrotic responses [83].

In a recent report using both human (*n* = 11) and murine (ICR 8–10 weeks) skin wound samples, IL-33 levels were high in both the inflammatory and proliferative phases. Double staining documented the presence of IL-33 in myofibroblasts during wound contraction. Importantly, the authors showed that IL-33 signaling through receptor ST2 is necessary for myofibroblast contraction [84].

***1.3D.*** 
**
*The Effects of CBD on IL-33*
**


That CBD could affect normal cutaneous wound healing should not be a surprise because the skin is rich with eCBs and their receptors, forming its own eCB signaling system. Most likely acting through TRPV (transient receptor potential cation channel subfamily V) channels and PPARγ (peroxisome proliferator-activated receptor γ receptors), CBD can alter gene expression through DNA methylation [85]. Others have confirmed a reduction in inflammation by day 3 in CBD-treated groups using a rat ventral tongue wound model [86]. This agrees with the current understanding of wound contraction due to smooth muscle activation by IL-33. The massive decrease in IL-33 level likely has other yet undefined roles in normal cutaneous wound healing. With time, this difference in healing rate disappeared with both groups healed at the same time.

**1.4** 
**Effects of cannabidiol on abnormal wound healing**


***1.4A.*** 
**
*CBD and Diabetic Wounds*
**


Pain carries a very negative connotation—we do many things and take multiple medications to avoid it. However, most believe pain is essential, without which noxious and often dangerous stimuli that can cause severe tissue injury will go undetected until it is too late. The patient with *FAAH* microdeletion demonstrated hypoalgesia, and indeed also has many scars due to injuries from normal daily activities from the lack of pain. Diabetic neuropathy causes a similar decrease in sensation which contributes greatly to injuries and wounds in these patients. The classic example of paresthesia-induced injury is the Charcot joint, described more than 140 years ago by Jean-Martin Charcot [87]. However, paradoxically, once wounds are present, pain is not uncommon, affecting 80% of patients with chronic wounds [88]. It is doubtful that at this stage pain remains helpful. Most of these wounds require daily dressing changes to remove the necrotic debris, keep the wounds clean, and isolate the wounds from the external environment. Such dressing changes can be painful. Unable to progress normally, these chronic wounds retain features of the inflammatory phase with high levels of neutrophils, M1 macrophages, and mediators such as TNFs, interferons, IL-1, IL-6, and matrix metalloproteinases (MMPs), among others. These dysregulations produce a sustained elevation in MMPs with insufficient inhibition by tissue inhibitors of metalloproteinases (TIMPs) and decrease the critically important growth factors in the wound, PDGF, TGFs, FGFs, EGF, and CTGF while disrupting their orchestrated, orderly presence. Recent reports indicate inflammasome activation in keratinocytes takes place by calcitonin gene-related peptide, CGRP, through the action of IL-1β in these chronic wounds [89]. Based on these data, it is quite possible that CBD, with its analgesic effects, and the inhibition of CGRP, IL-33 would be a good therapeutic candidate for these wounds. This is particularly important considering a recent placebo-controlled trial showing the efficacy of CBD in reducing lower extremity neuropathic pain [90].

Besides its anti-inflammatory effects, CBD shows a promising influence on skin components [91,92]. Through several molecular mechanisms, like the upregulation of heme oxygenase (HMOX1) and BTB (Bric-a-Brac, Tramtrack, and Broad) domains and the CNC (Cap’n’Collar) homolog 1 (BACH1) protein, CBD increases the proliferation of keratinocytes accelerating wound healing [91].

In addition to potentially beneficial effects in treating diabetic wounds, Lehmann’s group made an interesting observation that CBD treatment delayed pancreatic inflammation in non-obese diabetic mice [93].

***1.4B.*** 
**
*Wound Healing in CB2-Knockout Mice*
**


CB2 activation has been linked to M1 to M2 polarization and a general reduction in inflammation during cutaneous wound healing. This was confirmed using CB2 agonists JWH133 and GP1a [94]. However, recent experiments using CB1- and CB2-knockout mice did not agree with the results obtained by pharmacological manipulations of CB2. CB1 knockouts required a longer time to heal the same skin wounds than CB2-knockout mice, indicating a more important role for CB1 signaling during skin wound healing. Indeed, there were higher levels of inflammatory mediators such as Il-6, MCP-1, and TNF-alpha in the CB1−/− wounds. Interestingly, Il-1β and TNF-α co-stimulation of CB2−/− MSC still generated an increase in IL-6 release to 150 pg/mL, 5× compared to unstimulated CB2−/− MSC controls [93].

***1.4C.*** 
**
*The Role of CTGF*
**


Connective Tissue Growth Factor (CTGF), a multifunctional and dynamic protein, plays a pivotal role in orchestrating the cellular and molecular events during the healing process [73]. A CTGF acts as a central hub that integrates signals from various growth factors, cytokines, and extracellular matrix components. Its multifaceted interactions enable CTGFs to modulate cellular responses critical for the healing cascade. From stimulating fibroblast proliferation to influencing extracellular matrix production, CTGFs stand at the crossroads of cellular activities that drive tissue repair [95,96,97].

The extracellular matrix (ECM) serves as the architectural scaffold for tissue regeneration, and CTGF is a key player in its remodeling. By promoting the synthesis of ECM components such as collagen and fibronectin, a CTGF contributes to the formation of a supportive microenvironment for cell migration, adhesion, and tissue restructuring. This role is particularly crucial in wound healing and tissue regeneration [98].

CTGF’s influence extends to the realm of angiogenesis, the formation of new blood vessels. As an angiogenic factor, a CTGF facilitates the recruitment and proliferation of endothelial cells, fostering the development of a robust vascular network essential for nutrient and oxygen delivery to healing tissues [99,100]. The angiogenic properties of CTGFs are integral to their role in both physiological and pathological healing contexts.

Fibroblasts are pivotal in tissue repair, and CTGFs actively influence the differentiation of mesenchymal stem cells to differentiate into fibroblasts [101]. Also, this growth factor can activate fibroblasts and promote their differentiation into myofibroblasts, which contributes to wound contraction and the synthesis of contractile proteins, ensuring the mechanical integrity of healing tissues [102].

As early as 2015, using a wound model in non-human primates, Thomson et al. documented lower levels of intact CTGFs in diabetic baboons’ incisional wounds even though the CTGF mRNA levels were identical between diabetic and non-diabetic animals, indicating increased degradation in the diabetic wounds [103]. The two-fold increase at week 4 after wounding in TIMP-1 levels of non-diabetic baboons confirmed the low CTGF was not due to a lack of production but the result of excessive breakdown as the MMPs were not under sufficient inhibition.

Understanding the intricate role of CTGFs in the healing process opens avenues for therapeutic interventions. Targeting CTGF pathways could hold promise in modulating tissue repair dynamics, particularly in cases of impaired wound healing or fibrotic disorders. It is also important to better understand the influence of CBD on CTGF pathways and how it can be used in wound healing management since it is an easy and inexpensive treatment.

Our lab has been exploring the potential of CBD as a CTGF enhancer to fine-tune its activities, offering new possibilities for precision medicine in the context of tissue regeneration and repair.

## 2. Results

Our data show a near 70% decrease (from 26.6% to 8.33%) in the IL-33 level in dorsal cutaneous wounds in B6 mice given CBD compared to mice in the placebo group. There is an initial decrease in wound closure rate compared to control (Figure 9 and Figure 10).

We also document that the use of 10 mg of inhalant CBD contributes to an increase in the expression of CTGFs on the wound site in diabetic mice models (db/db) which may improve the healing process in diabetic patients (Figure 11).

Our results show a six-fold reduction in ILC-1 with CBD in the wild type but not the CB2−/− (Figure 12). ILC-2, on the other hand, shows a 2.5× increase with CBD comparing the CB2−/− to the wild type.

## 3. Discussion

Injuries and wounds are inevitable parts of life. Wound healing has been integral in our survival and thus, not surprisingly, well conserved during evolution. In the 650 million years of chordate existence, repeated innovations render this capability extremely complex and adaptive. At the macroscopic level, healing is an emergent process—requiring the interaction of many agents at the micro- and nano-levels, affecting, and affected by others. After injury, the wound undergoes a sequential transition from coagulation and the inflammatory phase, through proliferation, to remodeling, each phase with many agents and sub-components. Most of the time, the events progress in a well-coordinated manner and end with a nice, soft, strong, and functional scar. But as medicine and surgery continue to push our ability to care for sicker and older patients, and our anachronistic, paleolithic physiology evolved and adapted to an environment 10,000 to 100,000 years ago, we will encounter more chronic wounds and complications from them. Inflammation, a double-edged sword, is at the core of the problem and likely holds the key to the solution. The principal limitation is the need to take apart this complex system to investigate the roles and contribution(s) of the component or components. This is because for emergent phenomena in any complex adaptive system, such as wound healing, the very property of the system under investigation no longer exists when the system is taken apart.

Cannabidiol, a non-psychoactive phytocannabinoid, appears to have anti-inflammatory properties by reducing the wound levels of IL-33 in both experimental animals and patients undergoing surgery. In T2DM mice, db/db, CBD increased the levels of CTGF. Regarding the safety of CBD, more research and time are needed. However, in a recent meta-analysis based on 25 published reports, including 927 patients, there were limited adverse effects even using doses at 2000 mg/day, escalated from 5 mg/kg/day orally for treating refractory epilepsy. Topical and inhalational CBD administrations are at a lower dose, 25 mg/day to 150 mg/day and 400 μg/dose, respectively. At these dosages and for 1 to 12 weeks, they produced no demonstrable adverse effects [104]. There is insufficient data on the safety of CBD use during pregnancy.

## 4. Materials and Methods

The procedures and experimental model for wound induction and analysis are described in detail as follows: Briefly, mice were anesthetized using a gaseous mixture consisting of 30% oxygen, 70% N_2_O, and 2.5% isoflurane using a vaporizer. For maintenance of anesthesia, the isoflurane concentration was reduced to 1.5%. Mice breathed spontaneously via a breathing mask throughout the surgical procedure. Surgical instruments were autoclaved prior to initiating the procedure. During the procedure, for multiple animals, the instruments were sterilized using a bead sterilizer. We kept at least two sets of instruments available to alternate use between animals before touching them to tissue. Gloves, lab coats, or disposable gowns were always on during the whole procedure. A face mask, head cap, and safety goggles were used during surgery. The hair on the dorsal part of each mouse was removed before surgery. The dorsal part of each mouse was disinfected with iodophors, followed by a rinse with 75% (*v*/*v*) ethanol. This disinfection procedure was repeated 3 times alternating wipes of the iodophor and 75% (*v*/*v*) ethanol (alcohol). For optimal pain control, Buprenorphine SR was administered prior to the start of the procedure. A one-wound model was created using a sterile biopsy punch (Figure 10). The wound was created using a sterile disposable biopsy punch, 6 mm diameters with a plunger available commercially (BrainTree Scientific Inc., Braintree, MA 02185, USA). As post-operative management and care protocol, an analgesic agent (Buprenorphine SR, Edgewood Pharmacy, Warren, NJ 07059, USA) was administered once every 24 to 72 h for post-operative pain relief. All animals were individually caged (Single-Housed Animals) for the rest of the experimental time. The wounds were then treated with topical/inhalant application of CBD or vehicle control every other day until complete wound closure was achieved in 10–12 days. Animals were monitored twice daily for manifestations of pain and weight loss. To assess and monitor the healing process, all subjects were photographed using a digital camera at 4 periodic time points accordingly. Photographic images were compared, and differences were recorded. For flow cytometry analysis, fresh tissues from wound areas were placed in a tissue culture dish with 1 mL PBS+ 2% FCS, 2 mg/mL of collagenase type II, and 1 mg/mL of DNase type I for 30 min at 37 °C. Samples were then sieved through a cell strainer (BD Biosciences, Franklin Lakes, NJ 07417, USA), followed by centrifugation (252× *g*, 5 min, 4 °C) to prepare single-cell suspensions. Cells were then first incubated with antibodies against surface proteins (e.g., CTGF). After washing with PBS, the samples were then fixed and permeabilized using fix/perm concentrate before incubation with antibodies for intracellular factors and cytokines (e.g., IL-33). Samples were then washed and subjected to flow cytometry using NovoCyte Quanteun (Agilent Technologies, Santa Clara, CA 95051) and analyzed using FlowJo analytical software 10.10.0 Java Version: 17.0.8+7-LTS (BD Biosciences, Franklin Lakes, NJ 07417, USA). Isotype-matched controls were analyzed to set the appropriate gates for each sample.

To investigate the role of CB2 receptors after CBD administration in dorsal cutaneous wounds on the local innate lymphocyte (ILC) population, we performed wound healing experiments using CB2−/− and wild-type controls. For flow cytometry analysis of ILCs, a single-cell suspension was prepared from wound tissues as described previously [105]. In brief, samples of a single-cell suspension from wound tissues were sieved through a 100 µM cell strainer (BD Biosciences, San Diego, CA, USA), followed by centrifugation (252× *g*, 10 min) to prepare single-cell suspensions. The samples were incubated with CD45, lineage markers, and CD127. Next, cells were fixed and permeabilized using fix/perm concentrate (eBioScience, Thermo Fisher Scientific, Waltham, MA 02451, USA) before incubation with antibodies for intracellular staining with T-bet (ILC1s); Gata3 (ILC2s); RoRγt (ILC3s); and corresponding cytokines. Cells were then run through a 4-Laser LSR II flow cytometer. Cells were gated based on forward and side scatter properties and on marker combinations to select cells of interest. All acquired flow cytometry data were analyzed using FlowJo V10. Graphs and summary statistics were also used to assess the results. For statistical analysis, Brown–Forsythe and Welch ANOVA were used to establish significance (*p* < 0.05) among groups. For tissue quantification statistical analysis, we compared the area of expression in both placebo- and CBD-treated groups by using a two-way ANOVA followed by a post hoc Sidak test for multiple comparison (*p* < 0.05). For wound areas, a Mann–Whitney statistical test was performed using GraphPad Prism (Version 10.2.2).

## 5. Conclusions

The endocannabinoid system is an elaborate, complex, and adaptive monitoring and modulating apparatus. Phytocannabinoids mimic the actions of the endogenous bioactive lipids derived from arachidonic acid and have unequivocal and very wide-ranging effects, including decreasing inflammatory responses following cutaneous injuries. While we will continue to explore, at the macroscopic level, the therapeutic clinical applications, the efforts to understand mechanistically, at the micro- and nano-levels, why and how CBD causes these observed beneficial effects are increasingly more important. Such a multi- or trans-scalar (fractal) approach allows for the targeted expansion and refinement of CBD use.

## Figures and Tables

**Figure 1 ijms-25-07137-f001:**
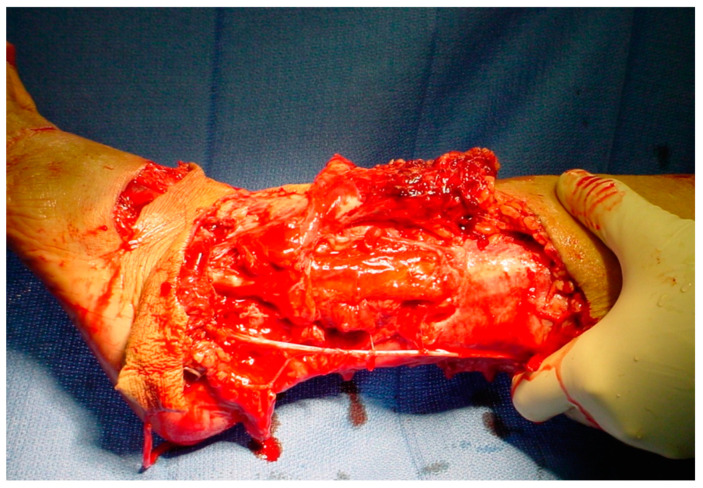
An avulsion, degloving wound of the lateral, distal left lower extremity with extensive destruction of the soft tissue.

**Figure 2 ijms-25-07137-f002:**
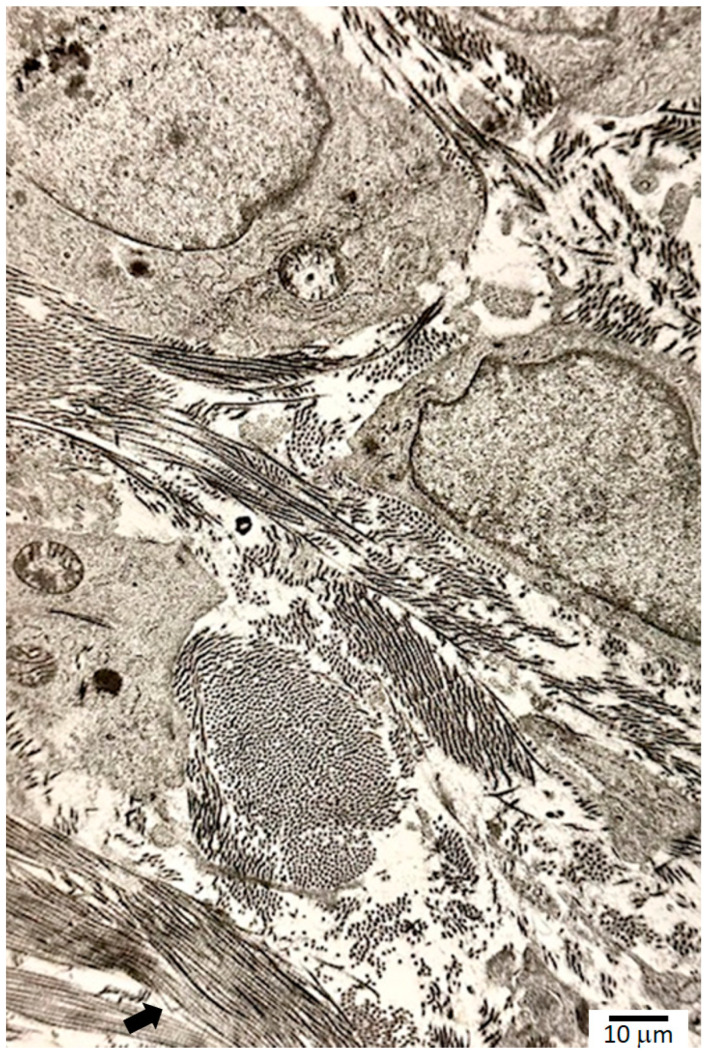
Transmission electron microscopy of fibroblasts and collagen, showing near-orthogonal fiber orientations. The 640 nm cross-banding pattern of the triple helix is visible in the lower left corner (arrow).

**Figure 3 ijms-25-07137-f003:**
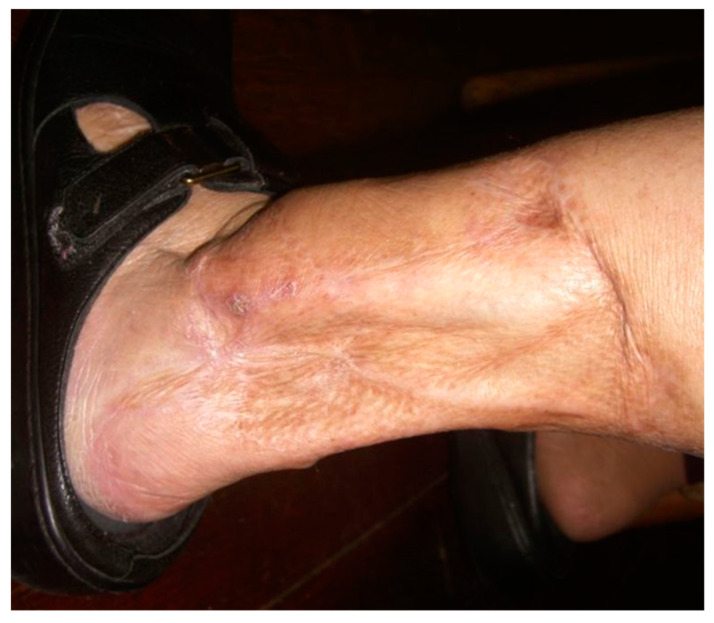
The above patient in Figure 1 with a healed distal left leg wound after debridement, with delayed and primary closures supplemented with split-thickness skin grafts.

**Figure 4 ijms-25-07137-f004:**
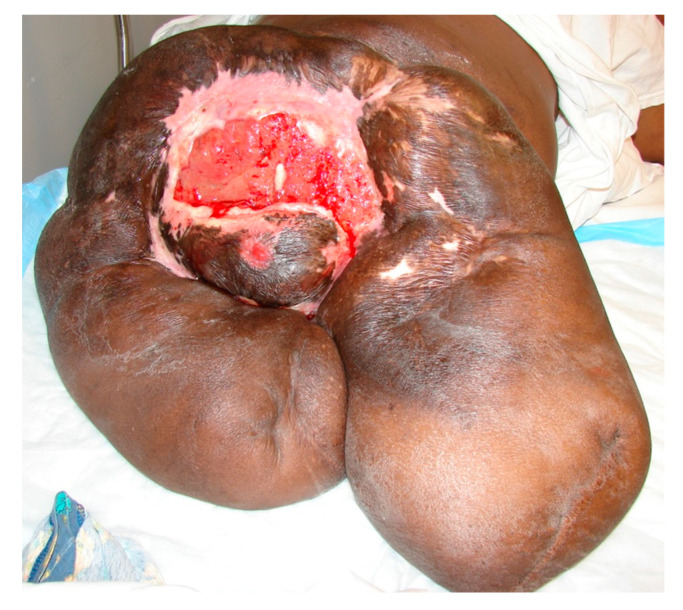
A complex chronic sacral, perineal wound in a paraplegic patient who already underwent hip disarticulation on the left side and above-the-knee amputation on the right side due to recalcitrant pressure ulcers. This wound eventually became Marjolijn’s ulcer with a malignant transformation into a highly aggressive squamous carcinoma.

**Figure 5 ijms-25-07137-f005:**
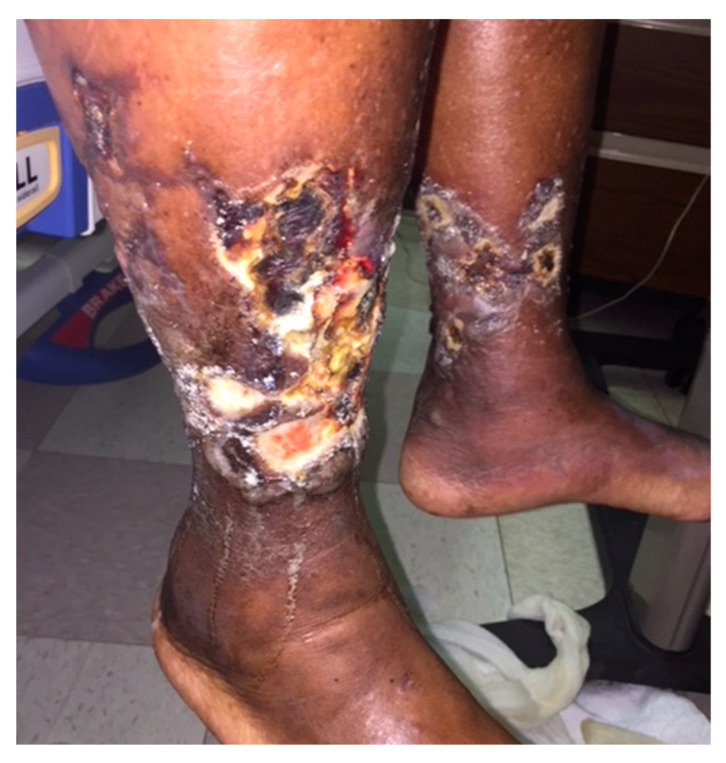
Chronic venous insufficiency causes extensive and prolonged ulceration in the gaiter region of both legs. This patient has a severe form of the condition and eventually succumbed to complications related to this.

**Figure 6 ijms-25-07137-f006:**
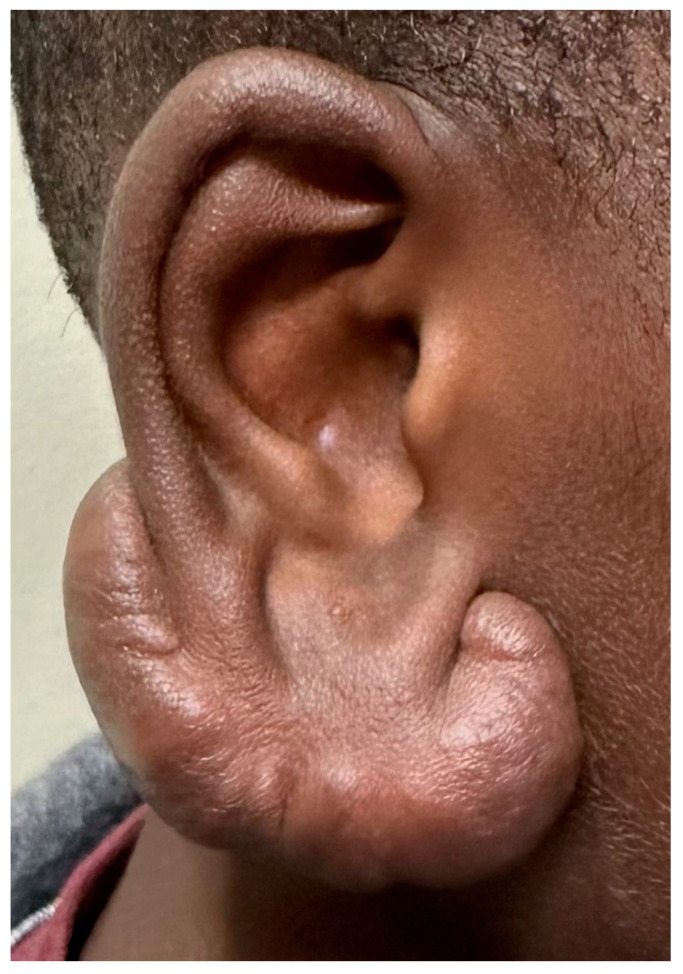
A large auricular keloid. It is firm, often with some continued enlargement and itching, and occasionally with increased warmth. In light of the Correia-Sá study, could this represent a deficiency in local arachidonoylethanolamine levels?

**Figure 7 ijms-25-07137-f007:**
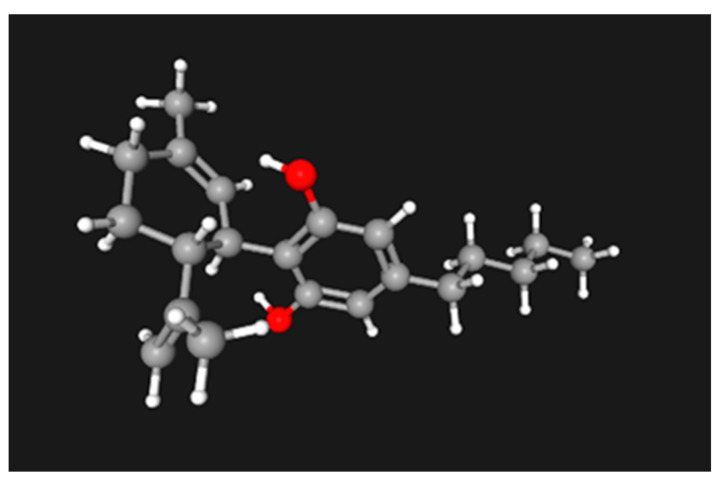
Cannabidiol molecular structure generated by 3-D modeler in PubChem.

**Figure 8 ijms-25-07137-f008:**
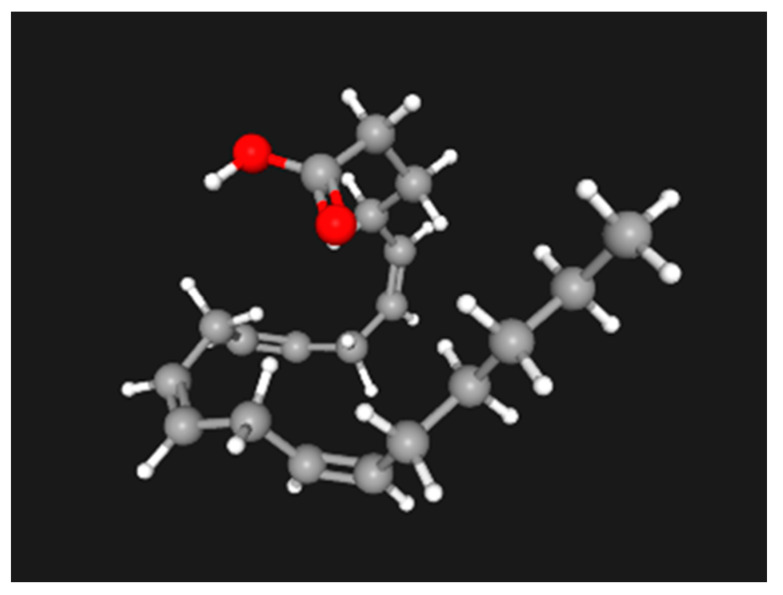
Arachidonic acid: icosa-cis 5,8,11,14-tetraenoic acid, the precursor for inflammatory mediators such prostaglandins, thromboxane, leukotriene, and two major endocannabinoids: 2-arachidonoylglycerol (2-AG) and *N*-arachidonoylethanolamine (anandamide, or AEA). The 3-D model was generated in PubChem.

**Figure 9 ijms-25-07137-f009:**
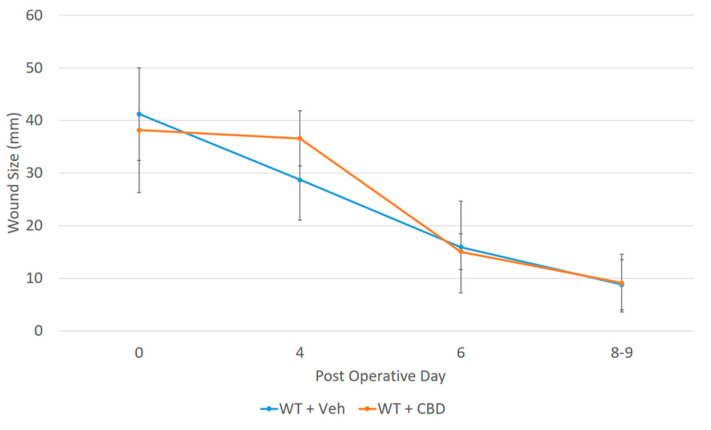
The alteration in the temporal wound healing profile by CBD showing a decrease in the initial rate of wound contraction, in agreement with the reduced levels of IL-33 in the wound. Error bars are standard deviations.

**Figure 10 ijms-25-07137-f010:**
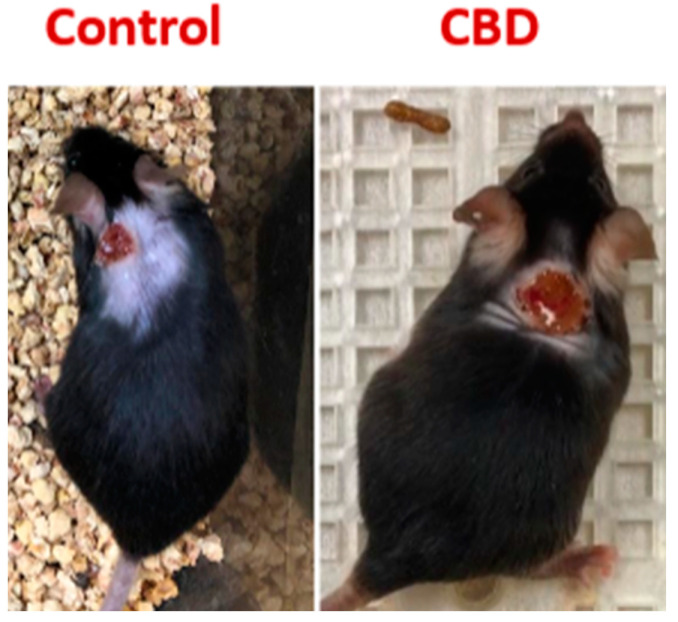
Examples of dorsal wounds in B6 mice showing initial delay in wound closure by contraction with CBD. Please note photographs are not taken with same magnification.

**Figure 11 ijms-25-07137-f011:**
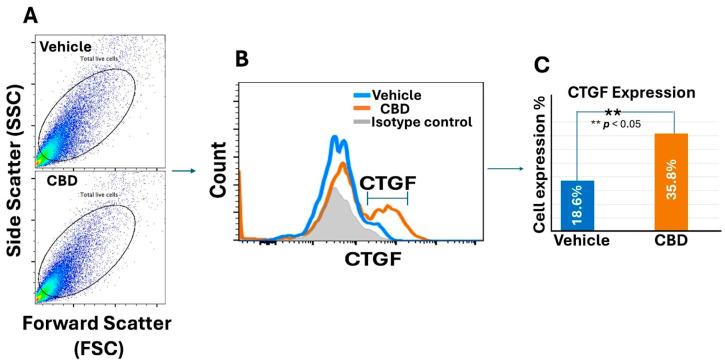
CBD-enhanced CTGF expression during wound healing process in db/db mice. (**A**) Flow cytometric panels (SSC/FSC), gated on live cells from wound sites treated with vehicle or CBD. (**B**) Histogram displays CTGF expression and (**C**) quantification of flow cytometry data (** *p* < 0.05).

**Figure 12 ijms-25-07137-f012:**
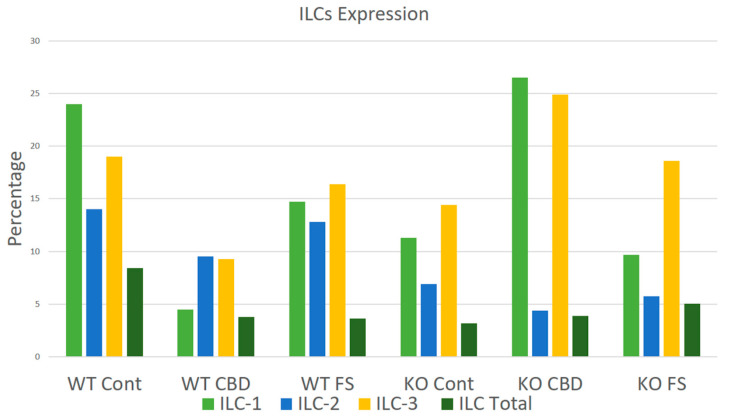
Changes in the ILC profiles of cells isolated from the dorsal wounds.

## Data Availability

Original data are available upon request, please send requests to the corresponding authors.

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
