# Peer review of "Cutaneous Wound Healing and the Effects of Cannabidiol"

_ijms, 2024, doi:10.3390/ijms25137137_

Round 1

Reviewer 1 Report

Comments and Suggestions for Authors

The objectives of this review are to provide clinicians engaged in wound care and basic science researchers interested in wound healing with an update, focusing on the potential roles of the endocannabinoid  system, cannabidiol, and the important immune-regulatory wound cytokine IL-33, a member of the  IL-1 family, due to its role in both normal and abnormal wound healing.

This review is very interesting but is necessary to implement some parts

About wound healing description

The description of wound healing is poor. There are several cellular elements that the authors forget to mention and describe. Is nececcary that the authors read some very recent review (2023) to enrich the review abou the organization of the cells

Same situations is for chronic wound

In summary the authors have to describe better the paragraphs about wound healing in both acute or chronic wounds using very recent review

Comments on the Quality of English Language

Moderate editing of English language required

Reviewer 2 Report

Comments and Suggestions for Authors

The manuscript examines the potential roles of the endocannabinoid system, particularly cannabidiol, and the immune-regulatory cytokine IL-33 in cutaneous wound healing. It highlights the need for a detailed mechanistic understanding, clinical comparisons, and thorough methodological descriptions to better contextualize cannabidiol's therapeutic potential in wound care. The study is of great importance and presents data well, but the manuscript requires major improvements to be suitable for publication. Pleaser find my comments below

Although the abstract is a review, mention any key findings or conclusions drawn from the review. This will provide readers with a quick snapshot of the main takeaways. The objectives of the study are somewhat vague. Explicitly state the research questions or hypotheses being tested. This will provide a clearer framework for the reader to understand the study's goals and significance.

The literature review should be expanded to include a comprehensive overview of existing research on the endocannabinoid system and cannabidiol in wound healing. This will help contextualize your study within the broader field and highlight the gaps your research aims to fill. For example, with the sentence ´Causal relationships in biology and medicine are often hard to estab-244 lish definitively, due to their multifactorial and context-dependent nature ´cite https://doi.org/10.4081/vl.2018.7196 to make literature up to date.

Provide a more detailed methodology section. Describe the specific experimental procedures, controls, and variables in detail. This should include the types of wounds studied, the dosages of cannabidiol used, and the methods of application. Add catalogue number of all materials and chemicals. Also, add all instruments, their model number supplier and software used.

Elaborate on the proposed mechanisms by which cannabidiol influences wound healing. Include detailed descriptions of the cellular and molecular pathways involved, supported by current research. If available, include clinical data or case studies to support the potential therapeutic benefits of cannabidiol in wound healing. This would strengthen the practical relevance of your findings. In Figure 2. Transmission electron microscopy, add scale bar in TEM micrograph and labels the 640 nm cross-banding pattern of the triple helix region for readers to understand.

Clearly describe the statistical methods used to analyze the data. This should include information on sample sizes, statistical tests performed, and measures of significance. Include a comparison of cannabidiol with existing standard treatments for wound healing. This will help readers understand the relative efficacy and advantages of cannabidiol. For example, in Figure 9, the alteration in temporal wound healing profile by CBD showing decrease in the initial rate of wound contraction, in agreement with the reduced levels of IL-33 in the wound, however, the error bars are not defined.  With the sentence ´ At the macroscopic level, healing is an emergent process – requiring the interaction of many agents at the micro and nano levels ´, cite https://doi.org/10.4081/vl.2018.7196 supporting the references.

Discuss the safety profile of cannabidiol, including any known side effects or contraindications. This is crucial for clinicians considering the use of cannabidiol in wound care. Provide a more detailed examination of the role of IL-33 in wound healing. Discuss its interactions with other cytokines and cellular pathways, and how cannabidiol may modulate these interactions. In Figure 11. CBD enhanced CTGF expression, the labels cannot be read. Also define gating criteria in scatter plot analysis showing figure.

Comments on the Quality of English Language

Minor editing of English language required

Round 2

Reviewer 1 Report

Comments and Suggestions for Authors

I reviewed the responses from the authors. The authors must excuse me but it is necessary to review some parts, essentially regarding acute and chronic wounds. In particular, regarding both acute and chronic wounds, the authors do not consider mast cells as fundamental cells in all phases of wound healing and in the failure of chronic wounds despite the fact that histamine and serotonin are considered. Furthermore, the role of angiogenesis and dendritic cells is missing. As regards chronic wounds, the same goes. The neuroimmunomodulation part must be expanded as the part dedicated to neuronal mediators is not complete.

Author Response

We have added more material and placed emphasis on the role of mast cells in normal and delayed wound healing, re-organizing the acute and chronic wound healing based on histology (epidermis, dermis, and subcutaneous fat) and cell types residing in them. Section 1B has also received additional discussion, specifically regarding neuronal responses and neuro-immunomodulation involving keratinocytes and dermal fibroblasts. Describing complex, multiple massively parallel processes such as wound healing using the single, sequential, line-by-line and word-by-word description has intrinsic limitations including blind spots. Your comments have been very helpful, allowing us to avoid omissions or simply listing a laundry list of events of what happens where and when, but rather on the how and why. Revisions are in green highlights, to help with the re-re-review.

Reviewer 2 Report

Comments and Suggestions for Authors

accept

Author Response

We thank the reviewer for their acceptance of the edits. 

Round 3

Reviewer 1 Report

Comments and Suggestions for Authors

I have appreciated the effort of the authors. They have asked correctly to my questions.

Comments on the Quality of English Language

Moderate editing of English language required